# Tolerance Induction in Cow’s Milk Allergic Children by Heated Cow’s Milk Protein: The iAGE Follow-Up Study

**DOI:** 10.3390/nu15051181

**Published:** 2023-02-27

**Authors:** Frank E. van Boven, Nicolette J. T. Arends, Aline B. Sprikkelman, Joyce A. M. Emons, Astrid I. Hendriks, Marloes van Splunter, Marco W. J. Schreurs, Severina Terlouw, Roy Gerth van Wijk, Harry J. Wichers, Huub F. J. Savelkoul, R. J. Joost van Neerven, Kasper A. Hettinga, Nicolette W. de Jong

**Affiliations:** 1Department of Internal Medicine, Section of Allergology & Clinical Immunology, Erasmus MC, University Medical Centre Rotterdam, 3015 GD Rotterdam, The Netherlands; 2Depertment of Peadiatric Allergology, Sophia Children Hospital, Erasmus MC, University Medical Centre Rotterdam, 3015 GD Rotterdam, The Netherlands; 3Department of Peadiatric Pulmonology & Allergology, University Medical Center Groningen, 9713 GZ Groningen, The Netherlands; 4University Medical Center Groningen, GRIAC Research Institute, University of Groningen, 9713 GZ Groningen, The Netherlands; 5Cell Biology & Immunology, Wageningen University & Research, 6708 PB Wageningen, The Netherlands; 6Department of Immunology, Erasmus MC, University Medical Centre Rotterdam, 3015 GD Rotterdam, The Netherlands; 7Food & Biobased Research, Wageningen University & Research, 6700 AA Wageningen, The Netherlands; 8FrieslandCampina, 3811 LP Amersfoort, The Netherlands; 9Dairy Science and Technology, Food Quality and Design Group, Wageningen University and Research, P.O. Box 17, 6700 AA Wageningen, The Netherlands

**Keywords:** cow’s milk, food allergy, heated protein, randomized controlled trial, safety, tolerance

## Abstract

Accelerating the induction of tolerance to cow’s milk (CM) reduces the burden of cow’s milk allergy (CMA). In this randomised controlled intervention study, we aimed to investigate the tolerance induction of a novel heated cow milk protein, the iAGE product, in 18 children with CMA (diagnosed by a paedriatric allergist). Children who tolerated the iAGE product were included. The treatment group (TG: *n* = 11; mean age 12.8 months, SD = 4.7) consumed the iAGE product daily with their own diet, and the control group (CG: *n* = 7; mean age 17.6 months, SD = 3.2) used an eHF without any milk consumption. In each group, 2 children had multiple food allergies. The follow-up procedures consisted of a double-blind placebo-controlled food challenge (DBPCFC) with CM t = 0, t = 1 (8 months), t = 2 (16 months), and t = 3 (24 months). At t = 1, eight (73%) of 11 children in the TG had a negative DBPCFC, versus four out of seven (57%) in the CG (BayesFactor = 0.61). At t = 3, nine of the 11 (82%) children in the TG and five of seven (71%) in the CG were tolerant (BayesFactor = 0.51). SIgE for CM reduced from a mean of 3.41 kU/L (SD = 5.63) in the TG to 1.24 kU/L (SD = 2.08) at the end of intervention, respectively a mean of 2.58 (SD = 3.32) in the CG to 0.63 kU/L (SD = 1.06). Product-related AEs were not reported. CM was successfully introduced in all children with negative DBPCFC. We found a standardised, well-defined heated CM protein powder that is safe for daily OIT treatment in a selected group of children with CMA. However, the benefits of inducing tolerance were not observed.

## 1. Introduction

Cow’s milk allergy (CMA) is a common disorder in Western countries, with a prevalence up to 3% in children [1,2]. Allergic reactions after consuming cow’s milk (CM) can be either IgE- or non-IgE-mediated [3]. These allergic reactions may range from mild dermal reactions to life-threatening anaphylaxis [2]. Tolerance is reached when the immune system no longer responds to an exposed antigen [4]. For children with IgE-associated CMA, approximately 57 to 87% achieve tolerance in the first year, with decreasing percentages in infants [1,5]. This percentage increases in subsequent years, but some children develop persistent CMA [6]. 

Complete avoidance of CM is the mainstay of CMA management until immunological tolerance is reached [7]. Other therapies include oral immunotherapy (OIT). OIT with baked milk products has shown promising results in inducing tolerance in a selected group of children [8]. In baked milk, the allergenic properties of milk proteins are altered by heat [9]. Baked milk products have been studied in various dietary products, such as cake, bread, and cheese on pizza [10]. However, the evidence remains limited [11]. Moreover, there is a need for more standardised products to overcome the heterogeneity of currently used dietary products.

The current study was preceded by a study from the same group in which the introduction of the standardised heated and glycated CM protein product (the iAGE-product) was studied for safety and tolerability in 25 children with challenge-proven CMA [12]. The iAGE product was tolerated by 72% of the 25 included children in the previous study. The tolerability of the product was associated with the outcomes of SPT with the iAGE product as well as the values of sIgE against α-lactalbumin, β-lactoglobulin, and caseins. The aim of the current follow-up study was to measure the tolerance-inducing effects of this new iAGE product in a group that initially tolerated the iAGE product, with a follow-up of 24 months.

## 2. Materials and Methods

### 2.1. Study Design

This study was a phase two randomized double-blinded, placebo-controlled trial with parallel groups. Eighteen children were randomly allocated to the treatment or control groups. The allocation list was produced using a random number generator program and was blinded until the end of the study.

### 2.2. Treatment Product

As described by de Jong et al. 2022 [12], Friesland Campina (Amersfoort, Netherlands) produced a heated and glycated CM protein powder (iAGE product). This product contained a mixture of casein protein (80%) and whey protein (20%) and was sterilised at 120 °C for 20 min. After heating, it was spray-dried and canned. To glycate the powder, the cans were stored at 60 °C for two weeks. The Quality Assurance/Quality Control (QA/QC) department of Friesland Campina approved the products and procedures for making the iAGE product compliant with the IFT guidelines. The amount of carboxymethyl lysine (CML) in the iAGE product was 300 ng CML/mg protein, which is comparable to that of evaporated milk.

The treatment product was dissolved in the children’s individual daily milk consumption (5% of total protein intake/day, depending on age) (Table 1). When a child entered a higher age group, the dose was increased, as shown in the table. Children in the control group received the same amount of extensively hydrolysed formula (eHF) by Friesland Campina (Amersfoort, Netherlands). Both groups were fed a cow milk-free diet. 

### 2.3. Patients

The study population in the preceding baseline study consisted of 25 children with challenge-proven CMA and was performed in seven Dutch hospitals specialized in pediatric allergy [12]. Parents of children between three months and three years of age were approached for participation in the study. The study population consisted of 9 girls and 16 boys (mean age 14.5 months; range: 6–36 months). 18 children used an extra hydrolysed formula (eHF) at the baseline visit, seven used an amino acid formula (AA). Four of the children had multiple food allergies (for instance egg, peanut, and nuts). SIgE values measured with ISAC against house dust mite, grass pollen and birch pollen were negative in all children [12]. After inclusion, all children underwent a double-blind placebo-controlled food challenge test (DBPCFC) with CM followed by clinical incremental introduction of the iAGE product [12]. In a previous study, 72% (18 out of 25) of the children with challenge-proven CMA could tolerate the iAGE product, asymptomatically. Consequently, 18 asymptomatic children were included in the follow-up study. Both parents and legal guardians of the children signed a written informed consent and were able to understand the Dutch language. The medical ethics committees of all seven involved hospitals approved the study protocol (NL61774.078.17, see Appendix A).

### 2.4. Primary Endpoint: DBPCFC Outcome

The children visited the hospital at 8 (t = 1), 16 (t = 2), and 24 months (t = 3) (Figure 1). During all follow-up visits, the patients underwent standardised (Table 2) DBPCFC with CM (primary outcome). DBPCFC was performed according to the guidelines described in a baseline study [12]. Symptoms were registered and the test was stopped using the PRACTALL criteria [13]. Daily treatment with the iAGE product was stopped when the child appeared to be tolerant of CM, as reflected by a negative DBPCFC test. Consequently, CM was introduced into the diet following a standardised introduction scheme supervised by a nurse.

Parents filled out several questionnaires at each follow-up visit (validated questionnaires as used in the “Generation R-study”, FAQLQ) [14,15]. Data were collected regarding atopy symptoms, Patient-Oriented Eczema Measure (POEM), eczema area and severity index (EASI) [16], dairy consumption in the past and present (breastfeeding, formula, soy), and adverse reactions in the past to CM.

The introduction of CM at home, which was advised after a negative challenge, was performed using an up-dosing schedule (see Appendix A) and monitored by the study nurse via scheduled (monthly) telephone calls. The highest dose of CM in the three weeks up-dosing scheme consisted of a daily minimum intake of 200 mL. When the child refused to drink CM, daily introduction to other dairy products such as yoghurt, was suggested.

### 2.5. Secondary Endpoint: Cow’s Milk Sensitization

During all visits, the skin prick test (SPT) with CM and the iAGE product was performed. The results were compared to the histamine equivalent prick (HEP) index score, as described previously [17]. In the absence of a cut-off level for the SPT-HEP, diameters of the SPT ≥ 3 mm Ø were considered positive [18]. 

Blood samples were obtained using either a finger prick (age < 6 months) or a venipuncture (≥6 months of age). Serum was collected and stored at −20 °C, and ImmunoCAP™ FEIA and ImmunoCAP™ ISAC (Thermo Fisher Scientific, Uppsala, Sweden) were used to detect CM extract (f2) sIgE and sIgE against the following cow’s milk components: α-lactalbumin (Bos d 4), β-lactoglobulin (Bos d 5), lactoferrin (Bos d 7), and whole casein (Bos d 8) [19]. sIgE > 0.35 kU/L, respectively, > 0.3 ISU were considered positive [20].

### 2.6. Adverse Events during the Intervention Period

Parents registered symptoms most likely caused by the use of the study product in a diary, including the daily amount of iAGE product added to the formula. Every month, a study nurse contacted the parents via telephone to evaluate possible symptoms and complaints. If necessary, the hospital physician was consulted. Events were reported on a data safety monitoring board (DSMB).

### 2.7. Statistical Analysis

This study was originally part of a large phase two randomized trial that aimed to test the efficacy of tolerance induction by the treatment product. As the trial was hampered by a small sample size due to a low inclusion rate, we decided to continue a pilot study with the included patients, focusing on measuring the safety and tolerance-inducing capability of the heated product. We analysed the data by calculating the Bayes factors (BF), which is the recommended method for evaluating small samples [21]. Bayes factors are defined as the likelihood ratio of the alternative hypothesis H1 (difference between groups) and null hypothesis H0 (no difference between groups). Evidence for the null hypothesis H0 was set as BF < 1, and evidence for the alternative hypothesis H1 was set as BF > 3 (moderate), BF > 10 (strong), BF > 30 (very strong), and BF > 100 (extremely strong) [21]. BF was calculated for differences in proportions of children from the treatment and control groups using the Savage-Dickey density ratio (as proposed by statistician Dr. J. Mulder, Correct usage/understanding of Bayes Factor when comparing two proportions, https://stats.stackexchange.com/q/457489). The priors in the proportions were estimated using a beta distribution with both parameters set to 1. All calculations were performed in R (version 4.1.2 (R Core Team (2022). R: Language and environment for statistical computing. R Foundation for Statistical Computing, Vienna, Austria. URL https://www.R-project.org/ (accessed on 1 February 2022)).

## 3. Results

### 3.1. Baseline Patient Characteristics

Eleven children (five female) were randomised to receive the iAGE product (treatment). The remaining seven children (one female) comprised the control group. The ages ranged from 6.5 to 19.4 months and from 13.4 to 22.5 months in the treatment and control groups, respectively. The parental report of eczema was comparable in both groups (64% respectively 57%), being in parity with positive scores on POEM; 55% respectively 57%. At baseline, no differences were found between the two groups (BF < 1.0) (Table 3), except for age < 12 months (BF = 3.4) and parental reports of asthmatic symptoms (BF = 4.3). In the latter two outcomes, the control group reported more observations. 

### 3.2. Primary Endpoint: DBPCFC with CM

At baseline (t = 0), 17 of the 18 children had positive DBPCFC with CM. One child was included after hospitalisation because of a severe CMA reaction prior to the start of the study. This child did not undergo DBPCFC with CM at baseline, since the diagnosis of CMA was confirmed with exposure at home. At t = 1, eight (73%) of the eleven children in the treatment group had a negative DBPCFC versus four out of seven (57%) in the control group. At t = 2, one extra child in the treatment group was tolerant compared with none in the control group. After 24 months (t = 3), nine of the eleven children (82%) in the treatment group had a negative DBPCFC, one was lost to follow-up between t = 0 and t = 1. In the control group, five of the seven children (71%) had a negative DBPCFC with CM, two were lost to follow-up between t = 0 and t = 2. The difference between the treatment and control groups was the largest at t = 1 (BF = 0.61), and smaller at t = 3 (BF= 0.51) (Figure 2).

### 3.3. Secondary Outcomes: SPT Results and sIgE to CM 

The mean SPT-HEP values for CM decreased in all children in the treatment and control groups from the start of the intervention (t = 0) to the end of the intervention (when DBPCFC was negative). The treatment group mean HEP 0.98 (range 0.00–3.75) at t = 0 decreased to 0.61 (range 0.00–2.84), and that of the control group 0.74 (range 0.00–2.58) at t = 0 also decreased to 0.27 (range 0.00–1.37) at the end of intervention. Specific IgE values for CM extract and components also decreased between the start and the end of intervention in all children: treatment group mean sIgE 3.41 kU/L (range 0.01–17.2 kU/L) decreased to 1.24 kU/L (range 0.02–6.44 kU/L) at the end of intervention, and control group mean sIgE 2.58 kU/L (range 0.0–6.87 kU/L) decreased to 0.63 (range 0.01–2.21 kU/L). The specific IgE values for allergen component (ISU) milk-specific allergens are shown in Table 4. The Bayes factors between the proportions of negative SPT and sIgE at the exit from the study did not differ for any of the outcomes (BF = 0.51–0.83).

### 3.4. Safety: Adverse Events during Treatment

Fifteen events were reported during treatment, none of which were related to the intake of the iAGE product (Table 5). In the treatment group, parents reported seven adverse events in four children. In the period from t = 0 to t = 1 (the first eight months of intervention), four of these events were probable allergic reactions. Two of the allergic reactions were likely late reactions due to DBPCFC when visiting the hospital at baseline (t = 0). The other two allergic reactions occurred in one patient, necessitating hospital visits (SAE). In both cases, the patient was treated with salbutamol. These latter two events were assessed (booked out) by peadiatricians as respiratory infections not related to the use of the iAGE product. Two other adverse reactions were also considered to be respiratory infections. Parents of children in the control group reported eight adverse events in four children during the period from t = 0 to t = 2, including four likely allergic reactions. These allergic reactions were most likely a late reaction due to DBPCFC when visiting the hospital at baseline (t = 0), as well as likely provoked by food challenge tests for CM and peanuts, and consumption of a deep-fried meatball. Six parents (Treatment 4, Control 2) did not complete the diary. Overall, the proportion of children with adverse events did not differ between the treatment and placebo groups (BF = 0.74). As the DSMB considered these events to be unrelated to the iAGE product, the patients were allowed to continue the study.

### 3.5. Compliance of Using the Treatment Product

In all cases, empty cans were returned to the hospital to confirm compliance. The 11 children in the treatment group consumed approximately 3360 doses of the iAGE product (total 3.4 kg). 

### 3.6. Introduction of CM after a Negative DBPCFC

Of the 14 children (one child was lost to follow-up) with a negative DBPCFC for CM (end of intervention, study exit), 13 successfully introduced CM proteins in their daily diet. However, this introduction did not cause severe adverse events. One child experienced mild gastrointestinal symptoms (stomach ache). Most parents succeeded in feeding the child the recommended minimum of 200 mL CM daily. In four cases, the child did not like the taste of CM. Three of them consumed cow milk-containing products daily, for example, biscuits, butter, cheese, pancakes, yoghurt (approximately 50 mL). One child consumed 100 mL chocolate milk daily.

## 4. Discussion

In this randomised controlled trial, we studied the safety and tolerance-inducing effects of heated cow milk protein powder (iAGE product) in children with CMA. Most of the children in both groups (treatment 73%, control 57%) became tolerant within eight months (t = 1, 8 months). At the end of the study period (t = 3, 24 months), 82% and 71% of children were tolerant to CM in the treatment and control groups, respectively, and CM was subsequently successfully introduced. Several adverse events were reported during the intervention period, but none was related to the study product, as assessed by paediatricians. The SPT results decreased for CM in both groups, whereas the SPT results with the iAGE product increased considerably in the treatment group and decreased in the control group. Positive sIgE values decreased in the treatment group for CM extract, and components Bos d 5, Bos d 4 and Bos d 7 sIgE levels were negative in all but one case before and in all cases after treatment. The SPT values in the control group also decreased over time for different CM proteins. Specific sIgE to Bos d 5 increased in one patient in the control group. Surprisingly, Bos d 8 levels at baseline in both groups were very low and consequently showed no room for improvement. Other serum sIgE levels in the control group followed the tendency of the treatment group. Overall, the treatment and control groups were comparable in terms of the primary and secondary study endpoints. 

The strength of the study was that, as far as we know, this was the first clinical study on the effectiveness of heated CM protein powder for the treatment of children with CMA. The most important reason for using this heated protein powder was the well-defined production method, for example, extensive sterilisation, exact period and temperature of the initial heating, as well as the glycation process. Additionally, the exact amount of CM protein offered to the children was known. Moreover, the powder can be easily added to the regular infant formula. It has no taste or colour which likely explains the high compliance observed in this study. This was an important benefit compared to the baked milk products used in previous studies, where the protein content in muffins and pizza was unknown [10]. 

This study was limited by its small sample size. Many studies on specific food allergies in children are underpowered [22]. Owing to the use of Bayes factors, we were still able to perform basic comparisons. Nevertheless, the Bayes approach did not allow for the detection of possible differences by treatment. In small samples this is even more challenging as the natural course in tolerance induction is already quite significant, leaving less room to improve. Therefore, meta-analyses are of importance to increase power and detecting effects by summarizing the results of small individual studies [22]. Our baseline comparisons showed significant differences in age and asthma symptoms. The sample size was insufficient to adjust for these imbalances. In the treatment group, five children younger than 12 months of age at baseline were included. Remarkably, all these children (100%) developed tolerance to CM within eight months (t = 1). Particularly, compared to cohorts showing decreased percentages of tolerance induction in infants of 50% to 57% in one to five years [1,5,23]. Unfortunately, we were not able to interpret this observation, as we could not compare it with the outcomes from the control group. Furthermore, at baseline, asthma-like symptoms were only present in the control group, but we had no indication that this affected our results.

Although a higher percentage of children seemed to become tolerant in the treatment group, particularly at t = 1, the numbers in our study were too low to find a trend. Consequently, we had to take into account that the percentages of tolerance in our study were influenced by the loss to follow-up of children in the treatment (*n* = 1) and control (*n* = 2) groups. One patient in the treatment group still had positive DBPCFC after 24 months. This patient, aged 20 months at baseline, was characterized by the highest skin prick test result with CM in our study group at t = 0 (SPT CM = 3.75 HEP; sIgE CM = 6.84 kU/l). High sensitisation profiles are predictive of persistent CMA [24].

The use of heated milk allergens for tolerance induction in children with allergies is of growing interest [25]. In a large study by Nowak-Wegrzyn et al., children tolerating a baked milk product outgrew their CMA earlier than patients who were intolerant to all milk products [26]. In a follow-up study, Kim et al. found that children tolerating baked milk products were more likely to become tolerant to unheated CM (OR = 2.8; *p* < 0.01) [8]. In addition to the use of a standardised well-defined product, another major difference between these studies and the iAGE study was the age at inclusion. Children in the iAGE study were included at 4 months—3 years, in contrast with the study by Nowak-Wegzryn et al.: mean age 7.5 years and the study by Kim et al.: age > 3 years [8,26]. We included younger children to speed up the normal tolerance induction period, which is for most CMA children before the age of 2 years [1]. The youngest children in our treatment group (<12 months at baseline; 5/11) were tolerant to cow’s milk after 8 months of using the study product. 

Kim et al. did not observe adverse events related to dietary baked milk during the treatment of 65 children who tolerated baked milk products at baseline [8]. In addition, a recent study by Nowak-Wegzryn et al. involving 136 treated patients also showed no adverse events related to the use of baked milk products at home [10]. The safety results of our study are consistent with those reported by Kim et al. and Nowak-Wegzryn et al. [8,10].

## 5. Conclusions

In conclusion, in this small study, we found that a standardised, well-defined, heated CM protein powder is safe for daily OIT treatment in a selected group of children with CMA. However, the benefits of inducing tolerance were not observed, most likely because of the small sample size. We recommend that future studies should test the effectiveness of tolerance induction acceleration using heated CM protein powder in infants with challenge-proven CMA in a larger sample.

## Figures and Tables

**Figure 1 nutrients-15-01181-f001:**
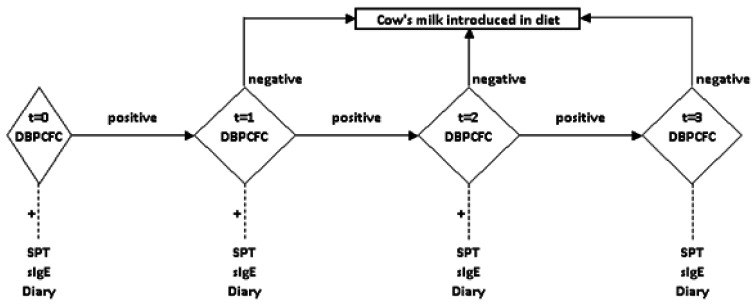
Flow chart iAGE study; DBPCFC: double-blind placebo-controlled food challenge; pos: positive; neg: negative; t: number of visits; SPT: skin prick test; sIgE: serum allergen-specific IgE.

**Figure 2 nutrients-15-01181-f002:**
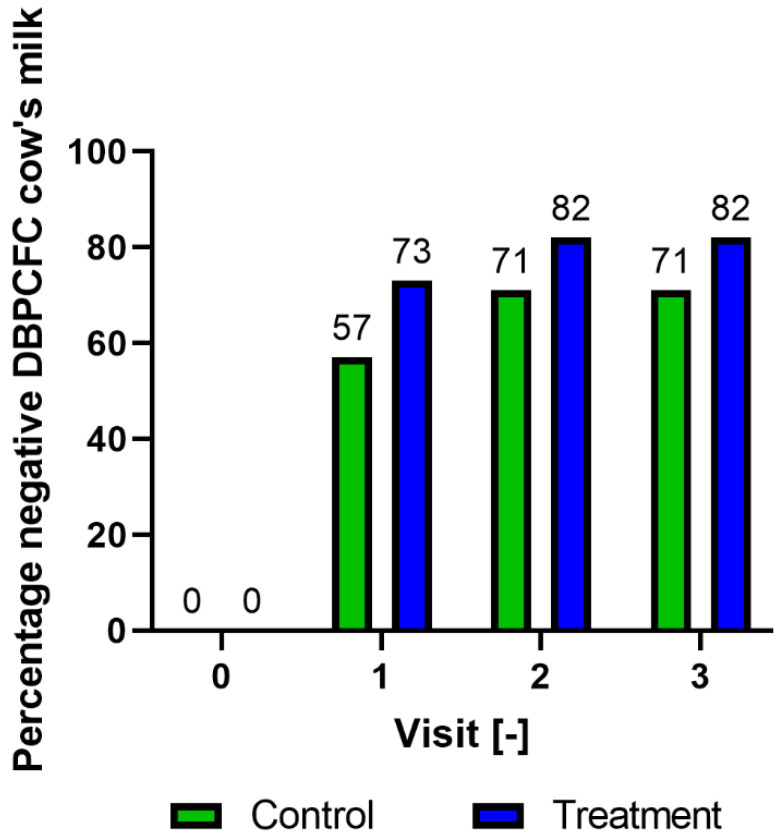
Results of the DBPCFC with cow’s milk; DBPCFC: double-blind placebo-controlled food challenge.

**Table 1 nutrients-15-01181-t001:** Daily dosages of the iAGE product during treatment.

Child		
Age (Months)	Weight (kg) ^1^	Protein via Study Amount (g)
3	6	0.8
6	7	0.8
9	8	0.8
12	9	1.1
24	10	1.1
18	11	1.1
21	12	1.4
24	13	1.4
30	14	1.4
36	15	1.7
48	17	1.7

^1^ kg = kilogram; g = gram.

**Table 2 nutrients-15-01181-t002:** Dosages in double-blind, placebo-controlled cow’s milk challenge, including the cumulative dosages.

Step	CM Protein (mg) ^1^	Cumulative Dosage (mg)
1	1	1
2	3	4
3	10	14
4	30	44
5	100	144
6	300	444
7	1000	1444
8	Age-dependent	Age-dependent

^1^ CM: cow’s milk; mg: milligrams.

**Table 3 nutrients-15-01181-t003:** Baseline characteristics of the 18 children participating in the iAGE-trial.

	Subject	Treatment				Control				
		Mean	Range	n_pos_ (Total)	%	Mean	Range	n_pos_ (Total)	%	Difference in Proportion Bayes Factor ^1^
	Age (months)	12.8	(6.5–19.4)	11		17.6	(13.4–22.5)	7		NA
	Age < 12 months			5(11)	45%			0(7)	0%	**3.4**
	Gender (F/total)			5(11)	45%			1(7)	14%	1.1
Atopy	Eczema			7(11)	64%			4(7)	57%	0.5
	EASI			3(11)	27%			2(7)	29%	0.5
	POEM			6(11)	55%			4(7)	57%	0.5
	Rhinitis			2(11)	18%			2(7)	29%	0.5
	Asthma-like symptoms			0(11)	0%			3(7)	43%	**4.3**
	Asthma + medication			0(11)	0%			2(7)	29%	
Exlusively breastfed	Period (month)	1.9	(0–8)	5(11)	45%	3.6	(0–9)	4(7)	57%	0.6
Formula use at inclusin visit	eHF			9(11)	82%			3(7)	43%	1.8
	AA			2(11)	18%			3(7)	43%	0.8
	AA/eHF			2(9)				3(3)		
Multiple food allergy	Egg, peanut and/or nuts			2(11)	18%			2(7)	29%	0.5
HEP	Cow’s milk	0.98	(0–3.75)	8(11)	73%	0.74	(0–2.58)	3(7)	43%	1.0
	Goat’s milk	1.06	(0–4.23)	5(9)	56%	0.40	(0–1.47)	2(7)	29%	0.9
	iAGE product	0.29	(0–0.84)	6(11)	55%	0.23	(0–1.58)	1(7)	14%	1.8

sIgE	Cow’s milk (kU/L)	3.41	(0.01–17.2)	6(9)	67%	2.58	(0–6.87)	4(6)	67%	0.5
	α-Lactalbumin Bos d4 (ISU)	0.24	(0–1.25)	3(9)	33%	0.67	(0–3.36)	2(6)	33%	0.5
	β-Lactoglobulin Bos d5 (ISU)		(0–10.72)	4(9)	44%	0.36	(0–2.13)	4(6)	67%	0.7
	Lactoferrin Bos d7 (ISU)			0(9)	0%			0(6)	0%	0.2
	Casein Bos d8 (ISU)	0.14	(0–0.96)	1(9)	11%	0.00	(0–0)	0(6)	0%	0.4

^1^ BF values in bold indicates evidence for the alternative hypothesis H_1_.

**Table 4 nutrients-15-01181-t004:** Sensitization to cow’s milk in SPT and sIgE at the onset of the intervention and at the end of intervention (negative DBPCFC = exit of study).

		Treatment Group (*n* = 11)		Control Group (*n* = 7)		
	Outcome	Start of Study *n*/Mean	Exit of Study *n*/Mean	Start of Study *n*/Mean	Exit of Study *n*/Mean	BF at Exit ^1^
SPT-HEP	CM	11/0.98	9/0.61	7/0.74	5/0.27	0.51
	iAGE product	11/0.29	9/0.81	7/0.23	5/0.04	0.51

SIgE	CM (kU/L)	9/3.41	9/1.24	6/2.58	4/0.63	0.83
	α-Lactalbumin Bos d4 (ISU)	9/0.24	8/0.25	6/0.67	4/0.0	0.61
	β-Lactoglobulin Bos d5 (ISU)	9/2.21	8/0.49	6/0.36	4/0.60	0.61
	Lactoferrin Bos d7 (ISU)	9/0.0	8/0.0	6/0.0	4/0.0	0.61
	Casein Bos d8 (ISU)	9/0.0	8/0.0	6/0.0	4/0.0	0.61

^1^ CM: cow’s milk; HEP: Histamine equivalent Prick; SPT: skin prick test (positive > 3 mm); IgE: immune globuline E (positive > 0.3 ISU; >0.35 kU/L); *n*: sample size.

**Table 5 nutrients-15-01181-t005:** Adverse events during the use of the treatment or control product.

Treatment										
ID	Age (Months)	Gender (M/F) ^1^	Multiple Food Allergy?	# Events	Week	Allergic Symptoms?	Possible Related to Treatment?	Symptoms	Medication	Remark
111001	17.3	F	Yes	1	29	Not likely	Not likely	Fever, coughing, green colored mucus	Use of paracetamol	
333003	8.9	M	No	0						
333004	12.9	F	No	1	1	Likely	Not likely	Skin rash, four days later cough	Use of neurofen	Possibly, the skin rash is a late reaction due to the provocations at baseline
333005	11.4	F	No	0						
333006	15.2	M	No	NA						
555001	8.9	M	No	2	13 17	Likely Likely	Not likely Not likely	Respiratory symptoms, first day feverDry cough and chest tightness (moderate)	Hospital, use of salbutamol/flixotideHospital, use of salbutamol	Assessed a respiratory infection Assessed a respiratory infection
555002	6.6	F	No	NA						
555003	6.5	F	No	0						
666003	19.2	M	No	NA						
888001	14.3	M	No	NA						
888003	19.4	M	Yes	3	1 36 37	Likely Not likely Not likely Not likely	Not likely Not likely Not likely Not likely	First day: soft feces Second/third day: fever, cough Fever, cough, inflammation of ears (severe) Fever, cough, inflammation of ears (severe)	Use of fluimicil Use of amoxiciline Use of amoxiciline	Possibly the soft feces are a late reaction due to the provocations at baseline
**Placebo**										
111003	13.4	M	Yes	NA						
333001	15.8	M	Yes	3	17 30 48	Not likely Likely Likely	Not likely Not likely Not likely	Fever, inflammation of ears (moderate) Unclear, possibly skin related Red skin, cough	Use of amoxiciline Use of cetrizine, skinoilUse of xyzal, cetrizine, salbutamol	After provocation to cow’s milk After provocation to peanut
333002	18.2	M	No	2	3 4	Not likely Not likely	Not likely Not likely	Fever, inflammation of ears, diarrhea , vomiting, chest tightness Continued symptoms of week 3	Use of amoxiciline Use of amoxiciline	
666001	22.5	F	No	2	1 10	Likely Likely	Not likely Not likely	Fever, cough, skin rash in face, stomac pain Symptoms of skin rash and bowel	Use of co-trimoxazol Use of of aerius	Possibly, the skin rash is a late reaction due to provocations at baseline After eating one deep fried meatbal
666002	16.2	M	No	1	9	Not likely	Not likely	Cough, inflammation of ears (severe)	Use of amoxiciline	
888002	15.9	M	No	NA						
999003	20.9	M	No	0						

^1^ M = male; F = female.

## Data Availability

The individual patient data of this study is not available due to privacy restrictions.

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
