# Peer review of "Tolerance Induction in Cow’s Milk Allergic Children by Heated Cow’s Milk Protein: The iAGE Follow-Up Study"

_nutrients, 2023, doi:10.3390/nu15051181_

Round 1

Reviewer 1 Report

Thank you for asking me to review the manuscript of van Boven et al. Overall the investigation of different protocols that might facilitate the tolerance induction in patients with food allergy is very interesting and useful. The current work is very interesting with appropriate references to support introduction and discussion.

Minor comments:

I would find the following reordering of the methods section helpful to follow the protocol used:

2.1 study design

2.2 treatment product

2.3 patients - here I would suggest a short descriptive text of the study population in order to have a stand-alone manuscript -besides reference 12.

2.4. Primary endpoint DBPCFC

2.5 Adverse events

2.7 statistical analysis

Kindest regards

Reviewer 2 Report

Thank you for the opportunity to review the manuscript titled, Tolerance induction in cow’s milk allergic children by cow’s milk protein: the iAGE follow-up study. In this abstract, the authors aimed to “investigate the tolerance induction of a novel heated cow milk protein, the iAGE product, in 18 children with CMA” (coped verbatim from Lines 26-27, and adjusted for grammatical fluency). While the study is interesting, the low numbers of participants, and the wide age range of those who did participate, limit the interpretation and thus, utility of the findings.

1. Abstract: Please add the mean± SD (or, median, IQR) of the children’s ages.

2. Abstract: Were these children diagnosed with cow’s milk allergy by an allergist? What were baseline and follow-up IgE levels?

3. Abstract: Did participating children have any other diagnosed allergies, or allergic comorbidities?

4. Methods, Section 2.1 Patients: The inclusion of children ages 3 months to 3 years is a possible limitation, given that about half of children will naturally develop tolerance to milk within a year (Schoemaker et al., Allergy, 2015).

5. Minor: Inconsistent use of abbreviations, such as POEM, which was introduced in both Lines 95 and 169).

6. Figure 2 is extremely difficult to read, until increased to 150%

7. Section 3.4, Safety Characteristics: While the limited number of adverse events is reassuring, it would be helpful to have provided demographic characteristics of those who did, vs. did not react.

Round 2

Reviewer 2 Report

Thank you for the opportunity to re-review this manuscript. The authors have addressed my original concerns. At this time, I have only two comments.

1. Some minor grammatical errors remains. For example, in Line 38: Product-related AEs… should read Product-related AEs (no apostrophe). These are minor issues, and can be handled at the editorial level.

2. Figure 2 is still difficult to read, and may benefit from landscape orientation.
